# Functional and Structural Properties of Interhemispheric Interaction between Bilateral Precentral Hand Motor Regions in a Top Wheelchair Racing Paralympian

**DOI:** 10.3390/brainsci13050715

**Published:** 2023-04-25

**Authors:** Tomoyo Morita, Hiromasa Takemura, Eiichi Naito

**Affiliations:** 1Center for Information and Neural Networks (CiNet), Advanced ICT Research Institute, National Institute of Information and Communications Technology (NICT), 2A6 1-4 Yamadaoka, Suita 565-0871, Osaka, Japan; morita@nict.go.jp (T.M.); htakemur@nict.go.jp (H.T.); 2Graduate School of Frontier Biosciences, Osaka University, 1-3 Yamadaoka, Suita 565-0871, Osaka, Japan; 3Division of Sensory and Cognitive Brain Mapping, Department of System Neuroscience, National Institute for Physiological Sciences, 38 Nishigonaka Myodaiji, Okazaki 444-8585, Aichi, Japan; 4The Graduate Institute for Advanced Studies, SOKENDAI, Shonan Village, Hayama 240-0193, Kanagawa, Japan

**Keywords:** hand movement, interhemispheric inhibition, magnetic resonance imaging, precentral motor region, wheelchair sports

## Abstract

Long-term motor training can cause functional and structural changes in the human brain. Assessing how the training of specific movements affects specific parts of the neural circuitry is essential to understand better the underlying mechanisms of motor training-induced plasticity in the human brain. We report a single-case neuroimaging study that investigated functional and structural properties in a professional athlete of wheelchair racing. As wheelchair racing requires bilateral synchronization of upper limb movements, we hypothesized that functional and structural properties of interhemispheric interactions in the central motor system might differ between the professional athlete and controls. Functional and diffusion magnetic resonance imaging (fMRI and dMRI) data were obtained from a top Paralympian (P1) in wheelchair racing. With 23 years of wheelchair racing training starting at age eight, she holds an exceptional competitive record. Furthermore, fMRI and dMRI data were collected from three other paraplegic participants (P2-P4) with long-term wheelchair sports training other than wheelchair racing and 37 able-bodied control volunteers. Based on the fMRI data analyses, P1 showed activation in the bilateral precentral hand sections and greater functional connectivity between these sections during a right-hand unimanual task. In contrast, other paraplegic participants and controls showed activation in the contralateral hemisphere and deactivation in the ipsilateral hemisphere. Moreover, dMRI data analysis revealed that P1 exhibited significantly lower mean diffusivity along the transcallosal pathway connecting the bilateral precentral motor regions than control participants, which was not observed in the other paraplegic participants. These results suggest that long-term training with bilaterally synchronized upper-limb movements may promote bilateral recruitment of the precentral hand sections. Such recruitment may affect the structural circuitry involved in the interhemispheric interaction between the bilateral precentral regions. This study provides valuable evidence of the extreme adaptability of the human brain.

## 1. Introduction

Over the last decades, neuroscientists have demonstrated that motor training causes functional and structural changes in the human brain [1,2,3,4,5]. In addition, there is a general interest in how training-induced motor plasticity observed in a controlled experimental setting can be seen in professional athletes and musicians who have experienced long-term training in specific movements [6,7,8,9,10,11].

Wheelchair racing is a popular parasport for athletes with spinal cord injuries or other disabilities. While there has been a strong interest in wheelchair racing in sports science [12], the plastic changes in the brain induced by long-term training in wheelchair racing remain unknown. The most important factor in achieving high performance in wheelchair racing is to perform left- and right-hand movements while maintaining an almost perfect synchronization; otherwise, the wheelchair will meander due to time lag between the left and right-hand movements, leading to time lost in the race. Hence, training in wheelchair racing provides a unique opportunity to train bilaterally synchronized upper limb movements, which are rarely experienced in everyday activities of normal life, as well as in other sports. Thus, examining the brains having long-term wheelchair racing training can be a good way to know how the brain changes due to long-term synchronized bimanual training. One could hypothesize that the interhemispheric interactions between the bilateral precentral motor regions are particularly well-trained through long-term wheelchair racing training.

In typically developed young adults, previous functional magnetic resonance imaging (fMRI) studies have reported that the ipsilateral (right) precentral hand section shows a decrease in blood oxygen level-dependent (BOLD) signals (i.e., deactivation) during simple unimanual (right-hand) motor tasks [13,14,15,16,17,18,19,20], which is thought to be associated with interhemispheric inhibition exerted from the contralateral (left) region [21,22,23]. In typical human development, interhemispheric inhibition is not present at birth but develops from childhood to adolescence [16,24]. Such interhemispheric inhibitory mechanisms may be important for enabling one hand control while suppressing the movement of the other. In everyday activities, there are many opportunities to independently control the left and right hands; thus, interhemispheric inhibitory interactions are normally well-trained. If interhemispheric interactions are affected by the motor activities experienced in everyday life, we may expect them to be affected by special training outside of daily activities. In the case of wheelchair-racing athletes, we hypothesized that inhibitory and facilitatory interactions must be well-trained owing to repetitive training of synchronized bimanual movements.

The long-term use of interhemispheric interactions in wheelchair racing training also leads to a hypothesis related to white matter microstructural properties. Diffusion-weighted MRI (dMRI) is the only available method to measure microstructural properties along white matter pathways in living brains. In brief, dMRI measures the diffusivity of water molecules restricted by white matter microstructure. This method allows us to identify trajectories of white matter tracts by tracking the orientation of diffusion signals (tractography) and evaluate microstructural properties in each voxel by quantifying the properties of diffusion signals. dMRI studies revealed that white matter is more plastic than previously thought by demonstrating that motor training causes microstructural changes in white matter tissue [4,5,25]. Furthermore, at the cellular level, recent neurobiological studies have shown that the plasticity of white matter myelin depends on neural activity. Such plasticity is considered one of the biological mechanisms for white matter plasticity reported in dMRI studies [25,26,27]. Hence, we expected that long-term wheelchair training would cause microstructural changes in the transcallosal pathway connecting the bilateral precentral motor regions. Such a pathway might frequently carry signals for interhemispheric interactions during bimanual movement training.

At present, combining fMRI and dMRI is the best approach to simultaneously evaluate functional and structural properties of interhemispheric communication from the same living human brains since each modality will provide response properties of bilateral precentral regions and microstructural properties of callosal fiber pathway connecting with them, respectively, with relatively higher spatial resolution. To date, an approach combining fMRI and dMRI provided many valuable insights into how the functional organization of the cerebral cortex is related to the properties of white matter tracts [28,29,30,31].

To examine the research questions regarding the relationship between wheelchair racing and interhemispheric communication, we performed a single-case neuroimaging study on an elite Paralympian wheelchair racer with paraplegia (P1). The participant began wheelchair racing at the age of eight years and had intensive long-term (23 years) training and exceptional competitive records. We acquired fMRI data during unimanual and bimanual tasks and dMRI data to investigate the functional and structural properties of the P1 motor system. We also obtained fMRI and dMRI data from three other paraplegic participants (P2–P4) who had long-term wheelchair sports (basketball and table tennis) training in which synchronized bimanual movements were not as critical as in wheelchair racing. The data from paraplegic participants were compared with those obtained from 37 able-bodied control volunteers. In the fMRI study, we tested whether P1 showed activation in the bilateral precentral hand sections and greater functional connectivity between these sections during a right-hand unimanual task compared with control participants. In the dMRI study, we primarily focused on the P1′s white matter microstructural properties of the transcallosal pathway connecting the bilateral precentral motor regions and compared them with control participants.

## 2. Materials and Methods

### 2.1. Participants

One participant with congenital paraplegia (P1) and three with acquired paraplegia (P2, P3, and P4) participated in this study. Importantly, they all had normal sensorimotor functions in the upper extremities on both sides. Details of the participants are presented in Table 1. P1 was an active, top wheelchair-racing Paralympian aged 30 years. P1 began racing at the age of eight years and has had long-term (23 years) wheelchair racing training since then. P1 won 19 medals in six consecutive summer Paralympic Games as of 2021. In contrast, P2, P3, and P4 were not top athletes at the level of P1, and none had received such long-term wheelchair racing training, although they had more than 30 years of leg nonuse and long-term wheelchair sports training than wheelchair racing (Table 1). P2, with paraplegia since age one, had 42 years of experience in wheelchair basketball and four years in wheelchair table tennis. P3, who became paraplegic at 17, had 27 years of experience in wheelchair table tennis and was a Paralympian in Río de Janeiro but won no medals. P4, who became paraplegic at 21, had 31 years of experience in wheelchair basketball, nine years of experience in wheelchair marathons, and nine years of experience in wheelchair fencing. P1, P3, and P4 had no somatic sensations (light touch and pinprick) in their lower limbs and were immobile. P2 had immobile lower limbs; however, there were some somatic sensations (light touch and pinprick). These were evaluated by a physiotherapist (N.K.) with more than ten years of experience. The participants’ handedness was confirmed using the Edinburgh Handedness Inventory [32]. P1, P3, and P4 were right-handed, whereas P2 was ambidextrous.

Regarding the control participants, we recruited able-bodied adults (n = 37; 37.4 ± 10.9 [mean ± standard deviation (SD)] years, range 25–59 years; 25 females). They had experience in various sports since school; however, none had participated in any particular sport. The handedness of the control participants was confirmed using the Edinburgh Handedness Inventory (95.9 ± 7.6). All control participants were right-handed. None of the participants had any history of neurological, psychiatric, or movement disorders.

The study protocol was approved by the Ethics Committee of the National Institute of Information and Communications Technology (NICT) and the MRI Safety Committee of the Center for Information and Neural Networks (CiNet; no. 2003260010). The details of the experiment were explained to each participant before the experiment, after which they provided written informed consent. This study was conducted in accordance with the principles and guidelines of the Declaration of Helsinki (1975).

### 2.2. General Procedure

In this study, fMRI and dMRI experiments were conducted. We first collected fMRI data using one scanner and then collected dMRI data using another scanner. We collected functional data from all four paraplegic participants and all 37 control participants when they performed right-hand and bimanual tasks. The dMRI data were collected from all paraplegic participants but only from 33 control participants. Findings obtained from functional data other than those reported in the present study have been previously reported [33,34]. fMRI and dMRI data were acquired using different 3.0-Tesla MRI scanners (fMRI, MAGNETON Trio Tim, Siemens, Germany; dMRI, Prisma, Siemens, Erlangen, Germany) because, in the institutional setup, the Trio Tim scanner has a broader space for peripheral devices used for motor experiments. In contrast, the Prisma scanner has a strong gradient coil (80 mT/m) suitable for dMRI data acquisition.

### 2.3. fMRI Experiment

Before the fMRI experiment, participants were given an overview of the tasks to be performed in the scanner and the chance to familiarize themselves with them through prior practice outside the MRI room. Afterwards, the participants entered the room and were placed in an MRI scanner. Their heads were immobilized using sponge cushions and adhesive tape, and their ears were plugged. The participants’ body parts (chest, pelvis, and shins) were fixed to the MRI bed using Velcro to reduce body movements during the tasks. When performing a task, participants were asked to close their eyes, relax their entire body, refrain from producing unnecessary movements, and think of the assigned task.

Each participant completed one experimental 160-s run for each task. The run comprised five task epochs, each lasting 15 s. Considering each epoch, the participants continuously exerted cyclic movements for each task in synchrony with the cyclic audio tones. The details of each task are described below. The task epochs were separated by 15-s baseline (rest) periods. Each run also included a 25-s baseline period before the start of the first epoch. During the experimental run, we provided participants with auditory instructions indicating the start of a task epoch (three, two, one, start). We also utilized a computer-generated ‘stop’ signal to notify participants when each epoch had ended. The participants heard the same cyclic audio tones but did not generate any movement during the rest periods. All the auditory stimuli were provided using MRI-compatible headphones. An experimenter (E.N.) standing beside the scanner bed checked whether the participants performed each task properly through visual inspection throughout the run.

### 2.4. Acquisition of MRI Data

#### 2.4.1. Functional MRI

fMRI images were acquired using T2*-weighted gradient echo-planar imaging (EPI) sequences using the Trio scanner and its 32-channel array coil at the NICT CiNet. A multiband imaging technique was used (multiband factor, 3) [35]. Each volume consisted of 48 slices (slice thickness: 3.0 mm) acquired in an interleaved manner covering the entire brain. The imaging parameters were as follows: repetition time [TR], 1000 ms; echo time [TE], 27 ms; flip angle, 60°; field of view, 192 × 192 mm^2^; matrix size, 64 × 64 pixels; voxel size, 3 mm × 3 mm × 3 mm. In total, 160 volumes were collected for each experimental run. A T1-weighted image was acquired with a magnetization-prepared rapid gradient echo (MP-RAGE) sequence using the Trio scanner for each participant. This image was used for normalization in the preprocessing of the fMRI data. The imaging parameters were as follows: TR, 1900 ms; TE, 2.48 ms; flip angle, 9°; field of view, 256 × 256 mm^2^; matrix size, 256 × 256 pixels; slice thickness, 1.0 mm; voxel size, 1 mm × 1 mm × 1 mm; contiguous transverse slices, 208.

#### 2.4.2. Diffusion MRI

The dMRI data were acquired using a single-shot spin-echo EPI sequence with the Prisma scanner and its 32-channel array coil. We used the multiband sequence provided by the Center for Magnetic Resonance Research, Department of Radiology, University of Minnesota (https://www.cmrr.umn.edu/multiband/ [36]). Each volume consisted of 78 slices (slice thickness, 2.0 mm) covering the entire brain. We acquired multi-shell dMRI data, including 100 diffusion-weighted images with three different b-values (six directions with b = 300 s/mm^2^, 30 directions with b = 1000 s/mm^2^, and 64 directions with b = 2000 s/mm^2^) and 12 non-diffusion-weighted b = 0 images (TR, 3300 ms; TE, 66 ms; flip angle, 90°; phase encoding direction, posterior-anterior; phase partial Fourier, 5/8; diffusion scheme, monopolar). The field of view was 256 × 256 mm^2,^ and the matrix size was 128 × 128 pixels. The voxel dimensions were 2 mm × 2 mm × 2 mm along the x-, y-, and z-axes. To correct the susceptibility-induced distortion of the EPI images in subsequent analyses, two b = 0 images with a reversed phase-encoding direction (anterior-posterior) were acquired at the beginning of the dMRI scan. A T1-weighted image was obtained with an MP-RAGE sequence using the Prisma scanner for each participant. This image was used as a reference in the dMRI data analysis. The imaging parameters were as follows: TR, 1900 ms; TE, 3.37 ms; flip angle, 9°; field of view, 256 × 256 mm^2^; matrix size, 256 × 256 pixels; slice thickness, 1.0 mm; voxel size, 1 mm × 1 mm × 1 mm; contiguous transverse slices, 208.

### 2.5. fMRI Task Design

We performed two tasks to assess cortical activation during hand movements (Figure 1A,B).

#### 2.5.1. Right-Hand Task

All participants continuously performed cyclic extension-flexion movements of their right wrist synchrony using 1-Hz cyclic tones. We prepared a device to control the range of wrist motion (Figure 1A) used in our previous study [33]. A movable handrest was mounted on the device, and the hand was fixed on the handrest, which indicated the wrist angle. Two stoppers were fixed onto the device to control the range of wrist motion across task epochs and participants. They were positioned to prevent the wrist from extending beyond a straight position (0°) and flexing beyond 60°. The participants had to touch one of the stoppers (0° or 60°) alternately with the handrest in synchronization with the 1-Hz audio tones while making controlled and continuous wrist extension–flexion movements.

#### 2.5.2. Bimanual Task

All participants continuously exerted cyclic extension-flexion movements of their left and right wrists in synchronization using 1-Hz tones (Figure 1B). This task was performed to confirm that the bimanual task activated bilateral precentral hand sections. The participants generated in-phase extension-flexion movements of both hands. The range of the wrist motion was between 0° and 60°, as shown in the right-hand task (see above). The range of motion was controlled using the same device for both left and right hands, as utilized in the right-hand task.

### 2.6. fMRI Data Analysis

#### 2.6.1. fMRI Data Preprocessing

To eliminate the effects of unsteady magnetization during the tasks, the first ten EPI images in each fMRI run were discarded. The imaging data were analyzed using SPM 12 (Wellcome Centre for Human Neuroimaging, London, UK) in MATLAB (MathWorks, Sherborn, MA, USA). The following preprocessing steps were performed for each participant. The default SPM parameters were used unless otherwise specified. First, all EPI images were aligned to the first EPI image of the first session using six degrees of freedom (translation and rotation about the *x*-, *y*-, and *z*-axes) for rigid displacement. Through this realignment procedure, we obtained data related to the position of the head that changed over time from the first frame, using six parameters. All participants had a maximum displacement of <1.5 mm in the x-, y-, or z-plane and less than 3° of angular rotation about each axis during each fMRI run. Therefore, we excluded no data from the analysis. The T1-weighted structural image of each participant was coregistered to the mean image of all realigned EPI images using affine transformation. Finally, the structural and realigned EPI images were spatially normalized to the standard stereotactic Montreal Neurological Institute (MNI) space [37]. Normalization parameters for aligning the structural image with the MNI template brain (tissue probability maps: TPM.nii) were calculated using the SPM12 normalization algorithm. The same parameters were used to transform the realigned EPI images. The normalized EPI images were resliced to a 2-mm isotropic resolution, and successful alignment was visually checked. Finally, the normalized images were spatially smoothed using a Gaussian kernel with a full width at half maximum of 4 mm along the *x*, *y*, and *z* axes.

#### 2.6.2. Defining the Precentral Hand Regions

We defined the regions of interest (ROIs) for the left and right precentral hand sections (left and right ROIs) in the MNI space based on data obtained from our previous study [38], in which we identified functional clusters in the left and right precentral hand sections when we vibrated the tendons of the left and right hands. We depicted the overlapping region between the functional clusters identified in the previous study [38] and the precentral regions in the Automated Anatomical Labeling (AAL) atlas [39]. The ROIs most likely covered the bilateral hand sections of the dorsal premotor (PMD) and primary motor (M1) cortices. In the following analyses, we searched for significant clusters of voxels within each of the left and right ROIs using a small volume correction (SVC [40]; *p* < 0.05, FWE-corrected for a voxel-cluster image generated at an uncorrected height threshold of *p* < 0.001).

#### 2.6.3. Single-Subject Analysis

Following preprocessing, we used a general linear model (GLM) ([41,42] to analyze the fMRI data. A design matrix is prepared for each participant. Considering this single-subject analysis, the design matrix contained a boxcar function for the task epoch in the run, convolved with a canonical hemodynamic response function. To correct the residual motion-related variance after realignment, six realignment parameters were included in the design matrix as regressors of no interest. In the analysis, global mean scaling was not performed to prevent inducing Type I errors in evaluating the negative BOLD responses (deactivation) [43]. We generated an image showing the task-related activity in each task for each participant, which was used in the subsequent analyses. In this image, the effect of cyclic tones was most likely eliminated because the participants heard the sound consistently during the task epochs and rest periods. In addition, we identified the significant activation and deactivation of the ROIs during each task for each paraplegic participant (Figure 1).

#### 2.6.4. Group Analysis

##### Brain Activation and Deactivation in the Left and Right ROIs in the Control Group

To identify the activation and deactivation of the ROIs during each task in the control group, we performed a second-level group analysis [44] using a one-sample *t*-test.

##### One-to-Many Two-Sample *t*-Test

We directly compared the task-related activity obtained from each paraplegic participant with the control participants to determine whether there were significant clusters of voxels within the left and right ROIs in which a paraplegic participant showed greater activity than the control participants. To assess statistical differences, we compared each paraplegic participant and controlled participants using the Crawford and Howell *t*-test, which is a modification of the regular independent two-sample *t*-test, allowing us to compare one sample with a group of multiple samples, where the within-group variance was estimated from the latter group by assuming that the within-group variances in the two groups were identical [45].

In this analysis, we included the age and sex of all participants as nuisance covariates (effect of no interest) because these factors could influence the evaluation of the present between-group differences. This approach has been widely used in GLM analyses to evaluate target factors by considering the effects of the nuisance covariates [33,46]. When a significant cluster was found in the right ROI, we extracted the activity parameter estimate from the cluster for each participant, plotted it, and calculated Cohen’s d to determine the effect size of the difference.

##### Functional Connectivity Analysis

We examined whether functional connectivity between the bilateral precentral hand sections during the right-hand task was greater in P1 than in control participants by performing a generalized psychophysiological interaction analysis (gPPI) [47]. We also tested this in the other three paraplegic participants. The CONN toolbox (CONN, 20.b; web.conn-toolbox.org) was used by inputting the pre-processed images into the toolbox.

We applied denoising using the component-based noise correction method (CompCor) in the toolbox [48] to eliminate physiological noise originating from white matter and cerebrospinal fluid. We also removed head-motion-related artifacts. The data were filtered using a band-pass filter (0.008 < f < 0.09 Hz) to remove very slow fluctuating signals such as scanner drift.

In the gPPI analysis, the left ROI was used as the seed region. Each participant’s time course of the average fMRI signal across the voxels in the left ROI was deconvolved using a canonical hemodynamic response function (physiological variable). Next, we performed a GLM model analysis using the design matrix. We included the following regressors: physiological variable, boxcar function for the task epoch (psychological variable), and multiplication of physiological and psychological variables (PPI). These variables were convolved with a canonical hemodynamic response function. Six realignment parameters were included in the design matrix as regressors of no interest. We generated images of voxels in which the activity changed with the PPI regressor for each participant to depict voxels in which the activity changed in association with the fluctuation of the left ROI activity during the task epoch. We directly compared the images obtained from each paraplegic participant with those obtained from the control participants using a one-to-many two-sample *t*-test (see above). Finally, we searched for significant clusters of voxels in the right ROI, generated a voxel-cluster image at an uncorrected height threshold of *p* < 0.001, and checked for substantial clusters in the ROI using the SVC. When a significant cluster in the right ROI was found, we extracted the parameter estimate of connectivity change from the cluster for each participant, plotted it, and calculated Cohen’s d to assess the effect size of the difference. In addition, we performed the same functional connectivity analysis for bimanual tasks.

### 2.7. dMRI Data Analysis

#### 2.7.1. dMRI Data Preprocessing

The dMRI data were preprocessed using the TOPUP and EDDY tools implemented in FSL (https://fsl.fmrib.ox.ac.uk/fsl/fslwiki) to correct for susceptibility-induced distortion, participant head motion, and eddy current artifacts [49,50]. The dMRI data were then aligned to T1-weighted images in the AC-PC space using 14-parameter constrained nonlinear co-registration implemented in Vistasoft (https://github.com/vistalab/vistasoft). The diffusion tensor model was then fitted to the dMRI data using a least-squares algorithm to estimate the fractional anisotropy (FA) and mean diffusivity (MD). FA and MD are representative indices for evaluating microstructural properties along white matter tracts and are widely used in dMRI studies (see [29,51]). FA and MD are computed from axial and radial diffusivity (AD) and radial diffusivity (RD). In a Appendix A, we also analyzed AD and RD better to understand the source of the differences between participants. Finally, we also used multi-shell and multi-tissue-constrained spherical deconvolution ([52]) to estimate the fiber orientation distribution in each voxel using MRTrix3 (https://www.mrtrix.org/ [53]) based on tissue segmentation performed on T1-weighted images.

#### 2.7.2. Tractography

We performed two types of tractography analyses (whole-brain tractography and ROI-to-ROI tracking) on the dMRI data to identify a tract of interest, a callosal pathway connecting the bilateral precentral motor regions.

Whole-brain tractography: We used whole-brain tractography to generate streamlines in the entire white matter of each participant and identified tracts of interest based on the ROIs (see below). We used constrained spherical deconvolution-based probabilistic tractography to generate streamlines from the dMRI data (iFOD2 algorithm implemented in MRTrix3; https://mrtrix.readthedocs.io/en/dev/reference/commands/tckgen.html [54]). The default parameters of the iFOD2 algorithm were used, except for the maximum streamline length (250 mm) and angle threshold. We used the Ensemble Tractography approach [55], in which five different angle thresholds (2.9°, 5.7°, 11.5°, 23.1°, and 47.2°) were used to generate 10 million streamlines (two million streamlines from each angle threshold) to reduce biases induced by parameter selection. We then removed the streamlines that did not predict diffusion signals using Linear Fascicle Evaluation [56]. The seed voxels used for streamline generation were randomly selected from the gray matter/white matter interface regions identified by tissue segmentation performed on the T1-weighted structural images [57]. Streamlines interfacing with cerebrospinal fluid (CSF) voxels, defined by tissue segmentation on T1-weighted images, were discarded during the tracking process to avoid contamination of CSF voxels in the MD analysis along the pathway. Using whole-brain streamlines, we first identified a callosal pathway connecting the bilateral precentral gyri as a tract of interest because of the strong hypothesis that pathways carrying signals between the bilateral precentral motor regions may be affected by long-term training in bimanual movement during wheelchair racing. To this end, we selected streamlines with endpoints within a close distance (4 mm) of the left and right precentral gyri identified from T1-weighted images of individual participants using the Desikan atlas implemented in FreeSurfer (https://surfer.nmr.mgh.harvard.edu/ [58,59]). We utilized a large Desikan-Killiany atlas ROI instead of the ROI of the hand region used in fMRI. Defining tract endpoints using a smaller ROI leads to difficulties reliably identifying this pathway, owing to reduced sensitivity. In addition, we identified the callosal pathway connecting the bilateral frontal cortex (forceps minor) from whole-brain streamlines as a control tract using the MATLAB Automated Fiber Quantification (AFQ) toolbox (https://github.com/yeatmanlab/AFQ [60]). In all participants, we successfully reconstructed the callosal pathway connecting the bilateral precentral gyri and forceps minor in both hemispheres.

ROI-to-ROI tractography: In addition to the aforementioned whole-brain tractography approach, ROI-to-ROI tracking was performed to evaluate how differences in microstructural properties could be observed in specific portions of the callosal pathway connecting the M1 hand regions. To this end, we generated streamlines connecting the bilateral M1 hand region ROIs, which were defined in MNI coordinates based on a previous study [38] (x = −33, y = −24, and z = 54 for left hemispheres; x = 33, y = −24, and z = 54 for right hemispheres). First, we selected voxels in the individual participant data corresponding to these MNI coordinates based on the transformation matrix obtained for spatial normalization between the individual data and the MNI template brain. Next, we placed an 8-mm radius sphere as the M1 hand region ROIs in each hemisphere. Finally, we performed CSD-based probabilistic tractography (iFOD2 in MRTrix3) to generate streamlines connecting the bilateral M1 hand region ROIs. We used default parameters for tracking, except for the maximum streamline length (250 mm). The streamline generation was repeated until 500 streamlines connecting the bilateral M1 hand regions were generated. This ROI-to-ROI tracking procedure was unsuccessful in three control participants because of difficulty in tracking crossing fiber regions between the callosal pathway and the corticospinal tract. Therefore, in a subsequent analysis of ROI-to-ROI tractography, we only focused on paraplegic participants (P1, P2, P3, and P4) and 30 control participants (37.1 ± 11.0 [mean ± SD] years old, range 25–59 years; 22 female).

For all identified tracts, we removed streamlines that met the following criteria as outliers: (1) streamline length ≥ 3 SD longer than the median streamline length in the tract; (2) streamline position ≥ 3 SD away from the median position of the tract [61].

#### 2.7.3. Evaluating Tract Microstructural Properties and Statistical Analysis

We used the MATLAB Automated Fiber Quantification (AFQ) toolbox [60] (https://github.com/yeatmanlab/AFQ) to evaluate the microstructural properties along the tracts of interest. Briefly, we resampled each streamline to 100 equidistant nodes. Microstructural properties (FA, MD, AD and RD) were calculated at each node of each streamline. The properties at each node were summarized by calculating the weighted average of the microstructural measurements on each streamline within that node. The weight of each streamline was based on the Mahalanobis distance from the tract core to minimize the partial volume effect of neighboring tissues and individual variabilities of the tract volume. We excluded the first and last ten nodes because they were susceptible to crossing with superficial white matter and partial voluming with gray matter. The results for the remaining 80 nodes were plotted as tract profiles.

For statistical analyses, we averaged the data of 80 nodes to obtain a single-number summary of the microstructural measurements (FA, MD, AD and RD) per participant. In addition, Cohen’s d was calculated to assess the effect size of the difference between each paraplegic participant and the control group. Finally, we also used the Crawford and Howell t-test to assess the statistical differences between each paraplegic participant and the control participants.

## 3. Results

### 3.1. Ipsilateral Precentral Activation during the Right-Hand Task in P1

In the right-hand task, P1 showed a significant cluster of active voxels in the right (ipsilateral) ROI (Figure 1C), which was not observed in the other paraplegic participants (P2–P4). All paraplegic participants showed a significant cluster of active voxels in the left (contralateral) ROI, similar to the control group (Figure 1C). In addition, all paraplegic participants, including P1, showed a significant cluster of deactivated voxels in the right (ipsilateral) ROI, similar to the control group (Figure 1C). In the bimanual task, all paraplegic participants showed significant clusters of active voxels in bilateral ROIs, as observed in the control group (Figure 1D). Hence, only P1 showed bilateral precentral activation during the right-handed unimanual task, as was observed in the bimanual task for all participants. In P1, the spatial position of the right cluster identified in the right-hand task overlapped with that observed in the bimanual task. The Dice coefficient of the spatial overlap of the right cluster between the right-hand and bimanual tasks was 0.69, suggesting the presence of an overlap.

When directly comparing the brain activity of each paraplegic participant with that of the control participants, only P1 showed a significant cluster of voxels (peak coordinates x, y, z = 34, −22, 60; 46 voxels; *p* < 0.001 corrected) with greater activity in the right ROI than in the control group (Figure 2A). This cluster was located in the anterior part of the right ROI, suggesting that the increased activity occurred mainly in the PMD. When we examined the data obtained from the identified cluster, P1 showed greater activity than the control participants. The value deviated by more than four SDs from the control mean (Figure 2B; d = 4.61). In contrast, the data obtained from the other paraplegic participants were within the range of the data obtained from the control participants.

### 3.2. Greater Functional Connectivity between Bilateral Precentral Hand Sections during the Right-Hand Task in P1

When functional connectivity during the right-hand task was examined, we found a significant cluster of voxels (peak coordinates x, y, z = 30, −8, 56; 14 voxels; *p* = 0.042 corrected) that showed greater connectivity with the left ROI activity within the right ROI in P1 than in the control group (Figure 3A). None of the other paraplegic participants exhibited greater connectivity. As observed in the one-to-many contrast analysis, the cluster was located in the anterior part of the right ROI, suggesting that enhanced activity coupling in the right precentral motor region occurred mainly in the PMD (Figure 3A). Upon examining the connectivity obtained from the identified cluster, P1 showed a greater value than the control participants, and the value deviated by more than three SDs from the control mean (Figure 3B; d = 3.71), whereas the data obtained from other paraplegic participants were within the range of the distribution of the data obtained from the control participants.

Furthermore, we analyzed the functional connectivity for the bimanual task; however, none of the paraplegic participants had a significant cluster of voxels showing greater connectivity in the right ROI when compared to the control group.

### 3.3. Microstructural Properties of a Callosal Pathway in P1

We analyzed the microstructural properties of the white matter pathways connecting the bilateral precentral motor regions using dMRI data. With this objective, we first identified white matter tracts connecting the bilateral precentral gyri using whole-brain tractography. Then, we evaluated all participants’ microstructural properties (MD and FA) along this pathway (see Materials and Methods). Figure 4A depicts the white matter tract connecting the bilateral precentral gyri (left panel) and the MD along this pathway (middle and right panels). We found that the MD of P1 was significantly lower than that of the control group; it deviated for two SDs from the control mean (Crawford and Howell *t*-test; d = 2.56, t32 = −2.52, *p* = 0.02). The other three paraplegic participants (P2–P4) did not show similar deviations (Figure 4A, middle and right panels). However, P1 was not significantly different from the control group when we evaluated the microstructural properties of this pathway using FA (Appendix A; d = −0.43, t32 = 0.42, *p* = 0.67; see Appendix A for AD and RD).

Next, we examined if the MD difference along the callosal pathway was specific to the tract connecting the bilateral precentral motor regions or prevalent across other parts of the corpus callosum. We also evaluated MD along the forceps minor (Figure 4B, left panel), a callosal pathway connecting the bilateral frontal cortices. However, there were no significant differences between P1 and the control group (Figure 4B, middle and right panel; d = 1.54, t32 = −1.51, *p* = 0.14), suggesting that the observed difference along the callosal pathway of the motor system might not be fully generalizable to other parts of the corpus callosum.

Finally, we performed a Appendix A (ROI-to-ROI tractography) by generating streamlines directly connecting the hand sections of the bilateral M1, which were estimated from a previous fMRI study (see [38] Materials and Methods). This analysis might have better sensitivity because of its specific focus on the hand section, but we could not identify this pathway connecting the bilateral lateral parts of the M1 in the three control participants due to the difficulty in tracking white matter regions where the callosal pathway and corticospinal tract intersect. Figure 4C depicts the white matter tract connecting the bilateral M1 hand sections in P1, identified using ROI-to-ROI tractography. We evaluated the MD along this pathway in all paraplegic and 30 control participants (Figure 4C, middle and right panels). We found that the MD of P1 was significantly lower than that of the control participants and deviated by more than three SDs from the control mean (Crawford and Howell *t*-test; d = 3.09, t29 = −3.04, *p* = 0.005). The other three paraplegic participants (P2–P4) did not show trends similar to those observed in P1.

## 4. Discussion

### 4.1. General

In the current study, we investigated how the functional and structural properties of the motor system of P1, who had received 23 years of wheelchair racing training, were specialized compared to those of control populations. P1 showed activation in the bilateral precentral hand sections during the right-hand unimanual task, as normally observed during the bimanual task, with greater functional connectivity between these sections than control participants. P1 also exhibited a significantly lower MD along the transcallosal pathway than control participants. These findings were not observed in other paraplegic participants.

One might argue that it is unclear why functional and structural properties of the motor system observed in P1 were not observed in other paraplegic participants, even though they had undergone long-term wheelchair sports training other than wheelchair racing. However, as hand sensorimotor functions were intact in all paraplegic participants, the factors of congenital or acquired paraplegia between P1 and the others could not explain these differences. Furthermore, the duration of training was not perfectly matched between P1 and the other paraplegic participants (Table 1). However, it is unlikely that the duration of wheelchair sports training is a main factor for the present functional and structural changes because P3 had spent more time on his wheelchair table tennis training than P1 spent on her wheelchair racing training. Therefore, one possible interpretation is that a type of trained movement (synchronized bimanual movement) may explain why functional and structural changes in the interhemispheric interaction were observed in P1 but not in other paraplegic participants. However, this interpretation can still be speculative due to a lack of longitudinal data to provide more direct evidence on the associations between the type of trained movement and observed functional and structural changes in the brain.

The movements performed during wheelchair racing have unique properties that need to be considered when interpreting the results in P1. It has been reported that if the timing and force of turning both wheels of a wheelchair do not match, the wheelchair will meander, thus slowing down the wheelchair race time, and such a movement is energy-consuming [62,63]. Therefore, it is very important to continue performing bilaterally synchronized upper limb movements when pedaling a wheelchair to speed up wheelchair racing time. P1 had received special training for the past 23 years. Other wheelchair sports, such as wheelchair basketball and wheelchair table tennis, do not always require precise bilaterally synchronized upper limb movements to achieve high performance. Hence, such long-term special training in P1 may have shaped her unique bilateral recruitment mode for the precentral hand sections, which is suitable for synchronizing bilateral upper limb movements. The greater functional connectivity between the bilateral precentral hand sections during the right-hand task in P1 suggests that these regions likely process the same sensorimotor information, even during right-hand movements, further supporting our view of the bilateral recruitment mode of the precentral hand sections in P1.

### 4.2. Ipsilateral Activation in the Right Precentral Motor Regions during a Right-Hand Task

In the present study, as in many previous studies, we confirmed ipsilateral deactivation in all participants during the right-hand task (see Introduction). However, in P1, ipsilateral deactivation was localized laterally in the right ROI, and activation was predominantly observed within the ROI (Figure 1C).

Interhemispheric connections between bilateral precentral motor regions can be facilitative or suppressive [64,65,66]. In the participants, except for P1, the net effect of the interhemispheric connection seemed to be inhibitory during the right-hand task (Figure 2). In contrast, in the case of P1, both the facilitated and suppressed sections coexisted in the ipsilateral precentral hand section, indicating a unique organization of the functional fields. This view was further corroborated when we examined brain activity using the right index finger during a simple button-pressing task (see Appendix A). Only P1 showed significant activation and deactivation in the ipsilateral precentral hand section, as observed in the right-hand task. Considering that interhemispheric inhibition between the bilateral precentral motor regions progresses from childhood to adolescence in typical human development [16,24], the special experience of P1, where bilaterally synchronized upper-limb movement (wheelchair racing) training began at the age of eight, might have affected the unique development of hand/finger representations in the ipsilateral precentral motor region. However, we cannot completely exclude the possibility that P1 has such a unique representation at birth.

Concerning the involvement of ipsilateral precentral motor regions in the control of the hand, previous non-human primate studies have shown that the cells in the precentral motor regions are involved in the control of the ipsilateral hand [67,68]. In addition, precentral motor regions can control the ipsilateral hand in the human brain [69]. Hence, there might be a possibility that the ipsilateral precentral activation is involved in the control of the right hand in P1. Finally, the facilitated section was mainly located in the PMD rather than the M1 (Figure 2A and Figure 3A). This was also the case when we examined the brain region where P1 showed greater activity than the control participants during the simple button-pressing task using the right index finger (Appendix A). Thus, interhemispheric facilitation between the bilateral precentral motor regions was likely to occur in the PMD during the right-hand task and the right-finger task.

Our previous study [33] reported activation of the foot section of M1 during a bimanual task in P1. We have proposed that such a rarely seen phenomenon, that is, a brain region that is not normally involved in certain information processing and is rather suppressed during it in able-bodied persons, is constantly disinhibited and ordinarily used for this information processing, could be better referred to as hyper-adaptation. According to this definition, the recruitment of the ipsilateral precentral hand section during the right-hand task in P1 (Figure 1C) can also be referred to as hyper-adaptation because activation in the right precentral hand section during the right-hand task is rarely seen in able-bodied young adults (Figure 2B), and P1 seems to ordinarily use this section in daily right-hand and finger movements (Figure 1 and Appendix A).

### 4.3. Development of Transcallosal Pathway in P1

P1 showed a significantly lower MD along the transcallosal pathway connecting the bilateral precentral hand regions than control participants (Figure 4), which was not observed in the other paraplegic participants (P2–P4). Furthermore, recent studies have demonstrated that motor learning [4,5,25] can induce white matter plasticity activity-dependently [26,27]. Hence, it is natural to interpret that long-term repetitive training of bilaterally synchronized upper limb movements in P1, which likely generated synchronized activity between the bilateral precentral hand regions, caused plasticity in the microstructural properties along this pathway.

A previous study [70] reported a relationship between the microstructural properties of the transcallosal pathway and instrumental musical training, which also requires bimanual skills. Specifically, Steele et al. (2013) demonstrated that early-trained musicians who began their training before the age of 7 years showed significant development of microstructural properties in the posterior midbody/isthmus of the corpus callosum when compared with late-trained musicians and non-musicians. In addition, they demonstrated a significant correlation between the microstructural properties of the corpus callosum and the age at the onset of musical training. These results suggest that the age of training onset can be an essential factor in how training of bimanual skills affects the microstructural properties of the transcallosal pathway. Considering this previous finding, we speculate that P1′s early onset age of training (age eight years) must significantly impact the microstructural properties of the interhemispheric pathway.

Previous studies comparing dMRI and histological data have shown that MD is negatively correlated with myelin volume and axonal volume fraction [71,72,73]. Hence, the lower MD along the transcallosal pathway in P1 may suggest a relatively greater myelin volume and/or axonal volume fraction than that in the other participants. This view is consistent with the idea that long-term training may cause plastic changes in microstructural properties along the transcallosal pathway of P1, and such plastic changes may enable abundant signal transmission between the two precentral motor regions.

One might ask why we only observed the effect in MD but not in FA. There are some possible explanations why MD may be more sensitive in identifying microstructural properties specific to P1. First, we consider that P1′s transcallosal pathway had a certain microstructural property to which MD is more sensitive. White matter tracts have several biological properties of the tissue microstructure, such as axon diameter and myelination, each impacting the conduction velocity [74]. While diffusion tensor metrics are not specific markers for each microstructural property [75,76], it is generally known that FA and MD do not always show the same trend [77,78], although both are highly reproducible metrics [79]. Therefore, FA and MD may have different levels of sensitivity to specific types of microstructural plasticity caused by long-term wheelchair racing training. In addition, we note that Appendix A showed that a large effect size observed in MD could not be fully explained by a specific effect on either AD or RD (Appendix A). There are some previous studies discussing that AD and RD are sensitive to different types of biological processes [80,81,82]. We speculate that a significant effect observed in MD can be explained by multiple factors rather than a specific factor solely affecting AD or RD. In the future, it will be essential to assess the microstructural differences observed in wheelchair-racing athletes by combining multiple structural MRI acquisitions to enable biophysical modeling of an observed MR signal in the white matter [83,84,85,86].

### 4.4. Limitations

While we observed large effect size, we cannot rule out the possibility that present results are not generalized to other wheelchair-racing athletes due to a limitation inherent to a single-case study. However, an individual who can show such outstanding performance with special long-term training cannot be found everywhere. In general, single-case studies are essential to unveil the characteristics of such individual brains and provide valuable insights into the extent to which the human brain can adaptively change [87]. Although we were able to recruit only one wheelchair racer in this study due to restrictions imposed by COVID-19, we believe that an extension of this work with a relatively larger number of wheelchair racers will provide important insight into what factors may affect plastic changes in functional and structural properties of precentral interhemispheric interaction.

## 5. Conclusions

Brain plasticity due to long-term training has been well-documented. However, a specific question of whether long-term training of wheelchair racing, i.e., synchronized bimanual training, would affect functional and structural properties of precentral interhemispheric interaction has not been examined. The current work aimed to address this specific research question. Our results were consistent with the idea that long-term synchronized bimanual training may change functional organization in the precentral hand sections and induce microstructural changes in the transcallosal pathway connecting bilateral precentral hand sections. The present study provided such novel evidence and advanced our knowledge of brain plasticity due to long-term physical training.

## Figures and Tables

**Figure 1 brainsci-13-00715-f001:**
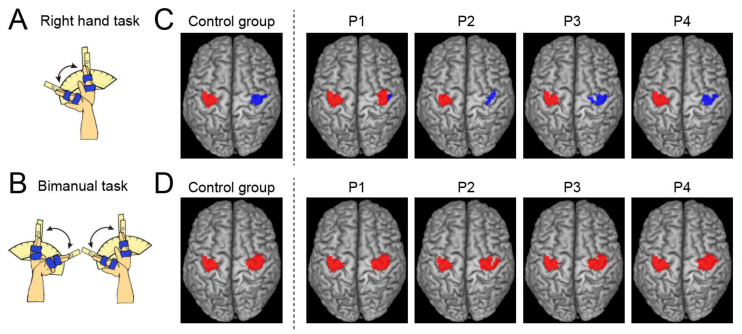
Tasks during fMRI experiment and results in group and single-subject analyses. (**A**,**B**): Schematic illustration of task used for the fMRI experiment ((**A**): right-hand task; (**B**): bimanual task). (**C**): Activation (red) and deactivation (blue) in the left and right ROIs obtained from the control group (leftmost panel) and each paraplegic participant (P1–P4) during the right-hand task. (**D**): Activations in the left and right ROIs were obtained from the control group (leftmost panel) and each paraplegic participant (P1–P4) during the bimanual task—abbreviations: ROI, region-of-interest.

**Figure 2 brainsci-13-00715-f002:**
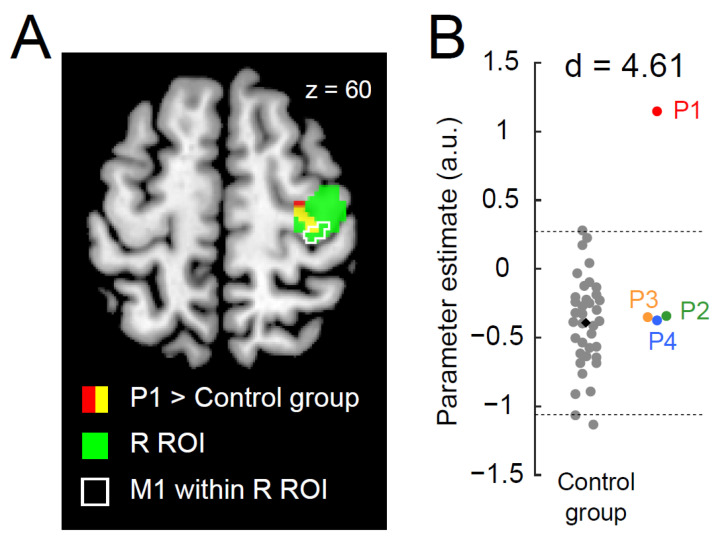
Greater activity in the right precentral hand section during the right-hand task in P1. (**A**): P1 demonstrates a significant cluster (red and yellow sections) of voxels showing greater activity than the control group. The green section indicates the right ROI. The yellow section indicates a significant cluster within the ROI. An area surrounded by white indicates the M1 (defined by cytoarchitectonic maps for areas 4a and 4p implemented in the JuBrain Anatomy toolbox, v. 3.0 [61]) within the right ROI. (**B**): Individual parameter estimates of the activity obtained from the identified cluster in the right ROI (yellow section in panel (**A**)). Each gray dot indicates the data obtained from each control participant, and a black dot indicates the mean across all control participants. Red, green, orange, and blue dots represent the data obtained from P1, P2, P3, and P4. Horizontal dashed lines indicate ±2 SD from the mean across the control participants. Cohen’s d value represents the difference in effect size between P1 and the control participants. Abbreviations: ROI, region-of-interest; M1, primary motor cortex; R, right; SD, standard deviation; a.u., arbitrary unit.

**Figure 3 brainsci-13-00715-f003:**
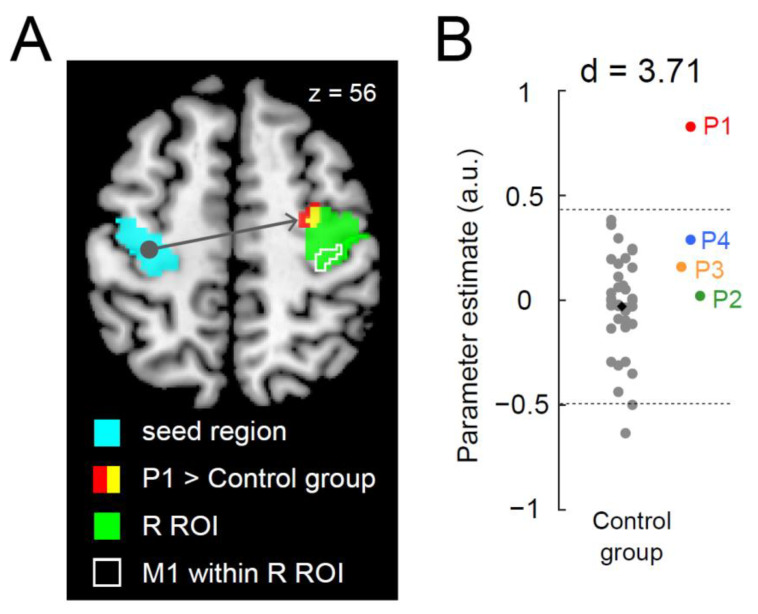
Greater functional connectivity in the right precentral hand section during the right-hand task in P1. (**A**): P1 demonstrates a significant cluster (red and yellow sections) of voxels showing greater functional connectivity than the control group. The light blue section indicates the left ROI (a seed region). The green section indicates the right ROI. The yellow section indicates a significant cluster within the ROI. An area surrounded by white indicates M1 (see Figure 2) within the right ROI. (**B**): Individual parameter estimates of the connectivity obtained from the identified cluster in the right ROI (yellow section in panel (**A**)). Cohen’s d value represents the difference in effect size between P1 and the control participants. The conventions are identical to those in Figure 2B. Abbreviations: ROI, region-of-interest; M1, primary motor cortex; R, right; SD, standard deviation; a.u., arbitrary unit.

**Figure 4 brainsci-13-00715-f004:**
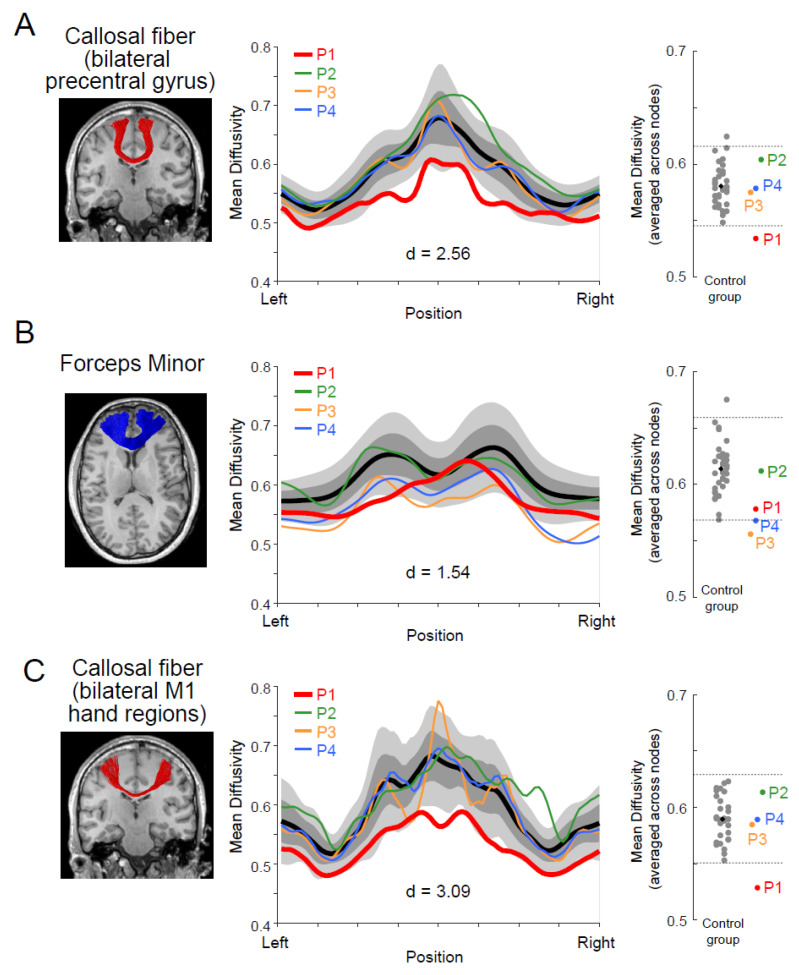
White matter microstructural properties (mean diffusivity; MD) of the callosal pathway in P1. (**A**): Left panel, a callosal pathway connecting the bilateral precentral gyri (red) of P1, identified by tractography, and overlaid on the coronal section of a T1-weighted image. Middle panel, tract profile along the pathway. Profiles of paraplegic participants are depicted as colored curves (red, P1; green, P2; orange, P3; blue, P4). The lighter gray shades indicate a range of ±2 SD from the control mean (thick black), and the darker gray band shows ±1 SD from the control mean. The horizontal axis describes the normalized position along the tract (left-right). The vertical axis describes MD. Right panel, MD along the pathway in each participant averaged across nodes. Each gray dot indicates the data obtained from each control participant, and a black dot indicates the mean across all control participants. Red, green, orange, and blue dots represent the data obtained from P1, P2, P3, and P4. Horizontal dashed lines indicate ±2 SD from the mean across the control participants. (**B**): Left panel, the forceps minor (blue), a callosal pathway connecting the bilateral frontal cortices of P1. Middle panel, tract profile along the pathway. Right panel, MD along the pathway in each participant averaged across nodes. Conventions are identical to those in panel (**A**). (**C**): Left panel, a callosal pathway connecting hand sections of bilateral M1 (red) of P1. Middle panel, tract profile along the pathway. Right panel, MD along the pathway in each participant averaged across nodes. Conventions are identical to those in panel (**A**). Cohen’s d value represents the difference in effect size between P1 and the control participants in each panel.

**Table 1 brainsci-13-00715-t001:** Participants’ information.

Participant	P1	P2	P3	P4
Age (in years)	30	61	54	52
Sex	F	M	M	M
Upper extremity motor subscores in ASIA	R:25/25 L:25/25	R:25/25 L:25/25	R:25/25 L:25/25	R:25/25 L:25/25
Upper extremity sensory subscores in ASIA	R:10/10 L:10/10	R:10/10 L:10/10	R:10/10 L:10/10	R:10/10 L:10/10
Leg non-use period (in years)	30	60	37	31
Neurological level	T12	Cannot be specified	T3	T8
Cause	* Spina bifida	Poliomyelitis	Spinal cord injury	Spinal cord injury
ASIA impairment scale	A	undefined	A	A
SCI	Complete	undefined	Complete	Complete
FIM	101/126	108/126	101/126	101/126
Main wheelchair sports (years played) Training period (age), training days/week, training hours/day	Track racing and marathon (23) 8–14 yo, 2 d/w, 8 h/d 15–17 yo, 7 d/w, 1–8 h/d 18–30 yo, 6 d/w, 2 h/d Total 17,600 h	Basketball (42) 17–22 yo, 4–5 d/w, 3 h/d 23–58 yo, 1–2 d/w, 3 h/d Total 12,150 h	Table tennis (27) 28–54 yo, 2–4 d/w, 2.5–8 h/d Total 20,250 h	Basketball (31) 22–35 yo, 5 d/w, 3 h/d 36–52 yo, 1 d/w, 3 h/d Total 13,050 h
Sub-wheelchair sports (years played) Training period (age), training days/week, training hours/day	None	Table tennis (4) 15–18 yo, 1 d/w, 2 h/d Total 300 h	None	Marathon (9) 27–35 yo, 2 d/w, 3 h/d Total 2700 h Fencing (9) 27–35 yo, 2 d/w, 3 h/d Total 2700 h
Handedness score	60	7	100	90

* Congenital. Abbreviations: ASIA, American Spinal Injury Association; SCI, Spinal Cord Injury; FIM, Functional Independence Measurement; T, thoracic; yo, years old; w: week; d, day; h, hours.

## Data Availability

Data supporting the findings of this study are available upon request from the corresponding author. The data are not publicly available because they contain information that can compromise the privacy of the study participants.

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
