# Peer review of "Functional and Structural Properties of Interhemispheric Interaction between Bilateral Precentral Hand Motor Regions in a Top Wheelchair Racing Paralympian"

_brainsci, 2023, doi:10.3390/brainsci13050715_

Round 1

Reviewer 1 Report

Comments and Suggestions for Authors

Introduction

1.       Lines 51-52: ‘…the wheelchair will meander, and time will be lost’. This part is not clear. Recommend rephrasing this.

2.       Authors stated that plastic changes in the brain induced by long-term training in wheelchair racing remain unknown. It is understandable but still it not convincing why it should be wheelchair racing. More rationales needed.

3.       Line 58-59: What do you mean by ‘deactivated’ during task?

4.       Lines 74-75: What do you mean by ‘white matter plasticity’? Need clarification.

5.       Line 76: What do you mean by ‘plasticity of white matter myelin’ and ‘activity-dependent’. It is very hard to understand without any background information.

6.       Line 77: What do you mean by ‘tissue changes’? It is very unclear. DTI measures microstructure of the brain using diffusivity of water molecules. Not sure what did the author mean by ‘tissue’. How it can be measured using dMRI?

7.       What is the rationale of using both fMRI and dMRI? I see the authors state the information of each measurement of neuroimaging modality, but it seems weak that why both should be measured. Also why dMRI, not volumetric measurement using T1-weighted MPRAGE?

Methods

1.       How the T1-weigheted structure MRI data were used as reference for fMRI and dMRI analyses given that fMRI and dMRI were collected from different scanners?

2.       Line 247-248: Was there a specific template used for the MNI space?

3.       I use AFNI to process fMRI data and haven’t used SPM, but I am wondering if SPM regresses out the signals from white matter and ventricles. Typically, the signals from those regions are regarded as physiological noise and they are regressed out during AFNI preprocessing.

4.       I am concerning about the ROI analysis. Since the fMRI data were normalized to the MNI space so it is technically okay to use ROIs based in the MNI space, but due to the variability of the brain sizes across individuals when they were normalized to the standardized space, using ROI analysis based on the template may not be very accurate. For this reason, I personally use FreeSurfer segmentation for the ROI analysis rather than ROI from the template. I would like to learn the author’s thoughts on this.

5.       Authors used FA and MD. Why not AD and RD were tested?

6.       dMRI data are single shell (b=1000). How did the authors use the single shell data for the multi-shell and multi-tissue-constrained spherical deconvolution using MRtrix? To my understanding, single-shell data don’t provide enough information regarding fiber crossing for the tractography analysis. I’m not sure if the single-shell data collected by the present study provides accurate information about tractography, so I’m not sure if the data are reliable. Also, what is the rationale of testing tractography analysis? It was not mentioned in introduction, but kind of appeared out of nowhere.

Results

1.       What is the cohen’s d appearing in Figure 1?

2.       What is the x, y, z coordinates for the cluster in Figure 1?

3.       In general, results are hard to follow and not straightforward. As the authors can see, the legend for each figure is very long. To me, it means that the data are not presented effectively. I’m not sure if readers can follow everything written there. Authors need to rethink how to present the data more effectively.

Discussion

1.       Line 655: Since the effects were observed in MD, not in FA, RD and AD should be analyzed for better understanding about the results?

2.       This sentence is not clear. ‘Because there is a theoretical limit to the maximum FA value (1.0), we speculate that FA was not a sensitive metric for evaluating the tissue properties of P1 along the pathway in which the control participants also showed generally higher FA values.’ As the author mentioned the average FA for control was 0.7 and still has a lot of room till 1.0 so this interpretation is not convincing.

3.       Overall this is an interesting study. My one comment is that the brain plasticity due to long-term training is very well-documented so it’s not that surprising to see that wheelchair racing induces brain plasticity using both fMRI and dMRI. Authors need to carefully think how to elaborate the necessity of the present study.

Reviewer 2 Report

Comments and Suggestions for Authors

The paper presents a single-case neuroimaging study with the aim to understand the underlying mechanisms of motor training-induced plasticity in the human brain.

The authors compared functional and structural properties of interhemispheric interaction between bilateral precentral hand motor regions in a top wheelchair racing Paralympian, and compared this pattern with control individuals.

The research topic is of relevance for both research and practice.

The manuscript is well written and understandable.

Important strengths include the detailed assessments (fMRI and dMRI).

The paper could nicely fit in the Special Issue “Physical Exercise-Driven Brain Plasticity”.

However, conclusions base on a single case (one person performing wheelchair racing) that is compared to three other paraplegic participants, and 37 able-bodied control volunteers. Can the pattern of results just have been observed by chance? At least this potential limitation needs to be discussed in more detail.

Only one person doing wheelchair racing is included. A bigger sample could allow more detailed insights.

The conceptual implications could be elaborated in more depth. How are current theoretical models advanced to better understand the underlying mechanisms of motor training-induced plasticity in the human brain?

The practical relevance could be better underlined with more detailed examples from real life.

Are findings relevant only in the high-performance sports context or also in habitual life?

Reviewer 3 Report

Comments and Suggestions for Authors

Thank you for the chance to read this very interesting and well-written manuscript.  Some minor suggested amendments are written on the attached draft of the manuscript.  

Author Response

Response: We appreciate the comments. We have changed as per the reviewer’s suggestion. Regarding the first point, we don't think the word “which” is necessary.